# Elongator Subunit 3 (Elp3) Is Required for Zebrafish Trunk Development

**DOI:** 10.3390/ijms21030925

**Published:** 2020-01-31

**Authors:** Diego Rojas-Benítez, Miguel L. Allende

**Affiliations:** FONDAP Center for Genome Regulation (CGR), Facultad de Ciencias, Universidad de Chile, Las Palmeras 3425, Ñuñoa, Santiago 7800003, Chile; mallende@uchile.cl

**Keywords:** elongator complex, tRNA modification, zebrafish

## Abstract

Transfer RNAs (tRNAs) are the most post-transcriptionally modified RNA species. Some of these modifications, especially the ones located in the anti-codon loop, are required for decoding capabilities of tRNAs. Such is the case for 5-methoxy-carbonyl-methyl-2-thio-uridine (mcm^5^s^2^U), synthetized by the Elongator complex. Mutants for its sub-units display pleiotropic phenotypes. In this paper, we analyze the role of *elp3* (Elongator catalytic sub-unit) in zebrafish development. We found that it is required for trunk development; *elp3* knock-down animals presented diminished levels of mcm^5^s^2^U and sonic hedgehog (Shh) signaling activity. Activation of this pathway was sufficient to revert the phenotype caused by *elp3* knockdown, indicating a functional relationship between Elongator and Shh through a yet unknown molecular mechanism.

## 1. Introduction

Most classes of RNA are post-transcriptionally modified in all cells. But transfer RNAs (tRNAs) are the most heavily modified: each one is decorated by a subset of over 90 known chemical modifications [1]. On average, ~14% of bases in a tRNA are post-transcriptionally modified [2]. Depending on their position, they play structural roles or are required for stabilizing cognate base pairing or expediting wobble base pairing, thus increasing decoding capabilities. As tRNAs, together with ribosomes, are the decoders of information held in mRNAs by recognition of consecutive triplets (codons) to add specific amino acids into the nascent polypeptide chains, modifications that affect the decoding capabilities of tRNAs will be of huge importance for translation. For instance, these types of modifications prevent frame-shifts, ensuring correct translation [3]. Mutations of tRNA-modifying enzymes display pleiotropic phenotypes in unicellular organisms and metazoans [3,4]. Many tRNA modifications are evolutionarily conserved, which has promoted the use of simple model organisms such as bacteria and yeast to understand the roles of modifications in cell physiology. Despite many years of research, there still remains much to elucidate in this field and the use of more complex models, including vertebrates, would advance our understanding of their roles.

We are interested in examining a well conserved modification, 5-methoxy-carbonyl-methyl-2-thio-uridine (mcm^5^s^2^U), which occurs at uridine in the wobble position (U_34_, Figure 1A) [5]. This modification improves decoding of A-ending codons GAA, CAA and AAA and for its synthesis, the six-unit Elongator Complex (Elp1-6) is fundamental as it adds carboxy-methyl (cm^5^) moieties to uridine bases at position 34. Elp3 is the catalytic subunit of the Elongator complex [5]. In humans, Elongator complex mutations have been related to neurological disorders [6,7]. Considering the scarce research of tRNA modifications in vertebrates, we aimed to use zebrafish and a morpholino-based approach to shed light onto the potential requirements of this modification during development. We found that morphant zebrafish showed diminished levels of the mcm^5^s^2^U compared to controls and showed defects in trunk development and muscle organization. The resulting phenotype correlated with diminished sonic hedgehog signaling (Shh) activity, consistently, activation of this signaling pathway was sufficient to revert the macroscopic phenotype caused by Elp3 knockdown.

## 2. Results

### 2.1. Knockdown of Elp3 Caused Diminished tRNA Modification and Generated Curved Larvae

Elongator complex is conserved through eukaryotes [8,9] and Elp3 is its catalytic sub-unit. To date, in zebrafish there is one ortholog annotated [10]. It is maternally deposited and by 27 h post-fertilization (hpf), its mRNA showed a non-spatially restricted expression pattern (Figure 1B), consistent with previous reports [10,11]. We wanted to establish if Elp3 function is conserved between yeast and zebrafish. We plated *ELP3* (Δ*ELP3*) mutant *Saccharomyces cerevisiae* alongside with control (BY4741), mutants that express either yeast *ELP3* (Δ*ELP3* + *ELP3*) or the zebrafish cognate (Δ*ELP3* + zElp3). In both cases, rescue of the slow growth phenotype was observed (Figure 1C), indicating functional conservation of the Elp3 function.

For *elp3* knockdown in zebrafish, we used two previously validated morpholinos (splice- and translation-blocking) [7]. We corroborated efficient silencing by detecting endogenous Elp3. Diminished levels of Elp3 protein were detected in morphants compared to control siblings (Figure 1D). To determine if the diminished Elp3 level impacted tRNA modification, we measured the levels of mcm^5^s^2^U in tRNAs from zebrafish using LC-MS (coupled liquid chromatography-mass spectrometry). In morphant animals, we detected a significant reduction in this modification compared to controls (Figure 1E), indicating a conserved molecular function for this gene in zebrafish. In morphants, we also consistently observed upregulated levels of the Unfolded Protein Response (UPR)-related genes BiP, Chop and Atf4 (Figure 1F). In morphological terms, we observed a consistent phenotype (65 ± 8% of injected larvae; *n* > 100) in morphants at 48 h post-fertilization (hpf), consisting of a ventrally-curved body. When the morpholino was co-injected with mRNA encoding for the yeast Elp3, rescue of the phenotype was observed (Figure 1G), indicating that the morpholino is specific and the observed phenotype is caused by diminished levels of Elp3.

### 2.2. Somite and Muscle Fiber Morphology is Affected Upon Elp3 Knockdown

Morphant embryos were consistently misshapen, displaying a downward curve of the body. As the larval body is mostly composed of developing muscle in the somites, we analyzed these structures by immunohistochemistry. To analyze somite shape, we detected β-dystroglycan, a protein present in extracellular matrix at myosepta as well as basement membrane between adjacent myofibers [12] that upon detection reveals the horizontal myoseptum and the borders of the somites. In morphant fish, the somites present a distortion of their typical chevron shape, they are rounded and show a different angle compared to control fish (Figure 2A,B). Morphants also show disruption of the horizontal myoseptum (Figure 2A, arrowheads). Furthermore, somite area is diminished in morphants compared to controls (Figure 2C). Although somite shape and size are abnormal, muscle fibers are correctly differentiated, as the typical striped-pattern of myomesin (Figure 2D) consistently, morphant embryos present muscle contraction (Figure 2E), indicating functional, contracting muscle although fibers appear disorganized.

We stained slow and fast fibers. For this purpose, we took lateral (Figure 3A–C,G–I) and cross-sections (Figure 3D–F,J–L) of 48 hpf embryos. Fast fibers appeared disorganized compared to controls (compare Figure 3A,B). In cross-sections, the trunk appeared misshapen in morphants (Figure 3D,E). Slow fibers also showed abnormalities, they appeared wrinkled in lateral views (Figure 3G,H), less dense and ordered in the cross-sections, lacking the side-by-side organization observed in controls (Figure 3J,K). In morphants, the neural tube appeared smaller than in controls (Figure 3E,K).

### 2.3. Elp3 Morphants Present Diminished Levels of Shh Pathway Transcriptional Targets

The ventrally-bent body phenotype in *elp3* morphant fish is reminiscent of a loss of function of sonic hedgehog signaling [13]. To determine if this signaling pathway activity is diminished in morphants and can be causally linked to the *elp3* knockdown phenotype, we performed immunofluorescence to detect the Engrailed 2a protein (En2a) and whole mount in situ hybridization to detect *col2a1* mRNA, both being transcriptional targets of Shh signaling [14]. We observed a significant reduction in the number of En2a+ cells (muscle pioneer cells) (Figure 4A,B) and diminished levels of Col2a1 mRNA levels in morphants compared to control siblings, suggesting that Shh activity is downregulated in the former.

### 2.4. Activation of Shh Signaling Rescues Elp3 Knockdown

Morphants for *elp3* present a ventrally bent phenotype and diminished levels of Shh signaling targets. To provide a functional relationship of *elp3* with the Shh pathway, we exposed control and morphant larvae to Purmorphamine (Pur), an agonist of Shh signaling which directly binds to Smoothened, activating the pathway [15]. In uninjected individuals, exposure to DMSO or Pur did not cause any morphological defects compared to control animals (Figure 5A–C). In the case of morpholino injected fish, the phenoptype was not altered by exposure to DMSO, but exposure to Pur reverted it (Figure 5D–F). To provide additional evidence for the interaction of *elp3* with the Shh pathway, we co-injected the Elp3-MO and the mRNA of a dominant negative form of cAMP-dependent protein kinase A (PKA^DN^), a negative regulator of Hedgehog signaling [16]. Figure 5G–H shows that the morphant phenotype was reverted by the presence of PKA^DN^, indicating that, at least part of the defect observed, is due to diminished Shh signaling. Together with the recovery of normal body shape, we also observed a reversion in the shape and muscle fiber organization observed in morphants (Figure 3). Furthermore, we observed recovery of En2a+ cells and Col2a1 mRNA levels (Figure 4). As control for mRNA injection, GFP mRNA was injected (Figure 5G), indicating that the observed rescue is specific to PKA^DN^ mRNA. Altogether, these results indicate that the observed phenotype in *elp3* morphants is caused, in part by diminished Shh activity.

## 3. Discussion

Mutants and knockdowns for tRNA-modifying enzymes present a plethora of phenotypes [3]. Elongator Complex (EC) has been shown to be involved in other cellular activities besides tRNA modification [17,18,19,20,21,22,23,24,25,26,27,28]. It is not clear whether these functions in which the complex is involved are consequences of impaired translation or are bona fide novel functions acquired during evolution. Something similar happens with the threonyl-carbamoyl transferase complex (TCTC, formerly known as EKC/KEOPS) as it is required for tRNA modification and also has been related with other functions [29,30]. Strong evidence in yeast support that the primary function of Elongator complex is tRNA modification, as mutation of Elp3 (Sin3p) in fission yeast leads to a severe reduction in the levels of mcm^5^s^2^U [31] and impaired growth could be reverted by the overexpression of two different tRNAs (tRNA_Lys_^UUU^ and tRNA_Leu_^UUG^), which are substrates for Elongator complex [32]. It is complex to dissect this in other organisms, as tRNA overexpression might not work as in yeast, considering the high numbers of genes which encode tRNAs in higher eukaryotes [33]. Also, mutant *C. elegans* [34], *Arabidopsis thaliana* [35], and mice [21] for Elongator subunits present reduced levels of mcm^5^s^2^U. In zebrafish, we found diminished levels of this modification in morphant zebrafish. Also, recent structural evidence strongly support the Elongator complex function is tRNA modification [36,37]. Altogether, this data strongly indicates that the primary and evolutionarily-conserved function of the Elongator complex is tRNA modification and that the other phenotypes observed in diverse species might be explained through defects in protein synthesis. Translation of multiple mRNAs is affected in absence of mcm^5^s^2^U in yeast and peripheral mammalian neurons [38,39], other organisms and cell types remain to be studied. Furthermore, not only the presence but also the levels of mcm^5^s^2^U-modified tRNAs seem to be a factor to consider, as not only knockouts must be studied as models, for example, in the case of IKAP (Elp1) to recapitulate Familial dysautonomia in mice, it was required to develop hypomorphic allele and not a complete knock-out [40], indicating that in some cases the relevant and realistic model, in terms of a pathological condition, might not be the lack of a particular modification, but rather diminished levels.

Most of what we know about the Elongator has been learnt by using unicellular organisms, especially yeast. The use of multicellular organisms has revealed tissue-specific requirements for tRNA modification as phenotypes are often restricted to a tissue or organ, most frequently neurons [22,39,40], though other cell types can be affected as we show here. We find a strong body shape malformation phenotype upon knockdown of the catalytic subunit of the Elongator complex, correlated with diminished levels of tRNA modification with mcm^5^s^2^U. We observed an upregulation of targets related to the unfolded protein response (UPR) which is consistent with previous data [25], suggesting that the synthesis of misfolded and unfolded proteins is occurring in this condition. This outcome has also been observed when other tRNA modifications are absent or their levels are diminished [25,41,42], supporting the idea that Elongator complex original function is tRNA modification. Previous work in humans and zebrafish had shown that the loss of *elp3* leads to a shortening of motoneuron axons [7]. We examined whether this was the case in *elp3* morphant zebrafish and only observed shortening in 5% of motor-neurons at 27 hpf, contrary to what Simpson et al. claimed using the same morpholinos we used [7]; also, by 48 hpf morphant showed normal motoneuron length (Appendix A). Furthermore, morphant embryos with no morphological aberrations, were able to move as much as control siblings (Figure 2E), indicating that motor-neurons are correctly innervating muscles, despite the disarray within the muscle fibers, these contract properly. Even though we show that *elp3* knockdown interferes with the sonic hedgehog pathway [43], motoneurons are not affected as in many other mutants in this pathway [44]. We also observed smaller eyes in *elp3* morphants, but there were no problems in establishing the left-right asymmetry as seen by the localization of the heart, indicating no problems in primary cilia formation or function [45]. *elp3* is expressed ubiquitously, suggesting that the modification is present in most tissues but, since the knockdown phenotype affects certain cell types, there may be differential requirement for the mcm^5^s^2^U modification. This has been shown in the case of t^6^A; mutants that disrupt this modification present cell type specific phenotypes [46,47]. In addition, in absence of Elongator, translation of several specific mRNAs is affected [48] that may explain the tissue- or cell-type specific phenotypes.

How translation is regulated in different contexts is of great interest and as tRNAs are central players in decoding genetic information, all forms of regulation over them may have a deep impact in gene expression. Post-transcriptional modification of tRNAs is highly dynamic and adaptable to different situations [49] and has been shown to have deep impact in protein synthesis [38,39,41,50,51]. Understanding the underlying mechanisms that regulate this process, might allow us to manipulate cell behaviors, cell fate or cell death and can have strong impact in cancer treatment [52,53,54] and even be important for establishment of long-term memory [55]. Deconvoluting and understanding tRNA modification absence is a hard task due of pleiotropic effects, nonetheless in the last decade methods for detection are actively improving [56] and in combination with high-throughput analyses (such as ribosome profiling and LC-MS) it will be possible to expand our knowledge on tRNA modifications role in cell physiology. The evidence so far and the ongoing research is strongly suggesting that tRNA modification changes could represent an additional layer of regulation of gene expression.

## 4. Materials and Methods

### 4.1. Zebrafish Maintenance

Zebrafish (*Danio rerio*) embryos were obtained by natural spawning. We used either TAB5 embryos as wild type. Embryos were staged according to Kimmel et al. [57]. We express embryonic and larval developmental times in hours post fertilization (hpf) or days post fertilization (dpf). Fertilized eggs were raised in Petri dishes at 28 °C containing E3 medium (5 mM NaCl, 0.17 mM KCl, 0.33 mM CaCl_2_, 0.3 mM MgSO_4_, and 0.1% methylene blue). All procedures complied with national guidelines of the Animal Use Ethics Committee of the University of Chile and the Bioethics Advisory Committee of Fondecyt-Conicyt (funding agency) project code 3160326 approved 26 October, 2015.

### 4.2. tRNA Modification Analysis

Control and morphants (48 hpf, 200 larvae each time) were collected and pooled from 3 independent experiments. For zebrafish total RNA extraction, the method by Rojas-Benítez et al. [46] was adapted. RNA was redissolved in nuclease-free water, store at −80 °C until shipment in dry ice to Arraystar (Rockville, MD, USA) for analysis as described in [39]. Peaks with a signal-to-noise ratio ≥5 were considered as detectable nucleosides. Peak areas were then normalized to the quantity of purified tRNA for each sample.

### 4.3. Quantitative PCR (qPCR)

For total RNA extraction Trizol (Life Technologies, Carlsbad, CA, USA) was used. RNA was extracted from pools of 50 embryos either control or morphants. cDNA was retro-transcribed from 1 µg total RNA with oligo (dT) primer in a 20 µL reaction volume using Improm-II Reverse Transcription System (Promega, Madison, WI, USA) and diluted to 50 µL with nuclease-free water. Real time PCR was set up using 2 µL cDNA, 10 µL SYBR Green Master Mix (Agilent Technologies, West Cedar creek, TX, USA) and 250 nM of each forward and reverse primers in a total volume of 20 µL. The qPCR was run for 40 cycles in a Stratagene Mx3000P thermocycler (Agilent Technologies, Waldbronn, Germany). GAPDH (F: TGACCCATTCATTGACCTTG, R: GCATGACCATCAATGACCAG), Atf4 (F: TTACGCATTGCTCCGATAGC, R: GCTGCGGTTTTATTCTGCTC), BiP (F: ATCAGATCTGGCCAAAATGC, R: CCACGTATGACGGAGTGATG) and Chop (F: ATATACTGGGCTCCGACACG, R: GATGAGGTGTTCTCCGTGGT). We used GAPDH for normalization, and the relative quantification of gene expression was calculated using the Pfaffl method. The data was displayed as a fold change of morphants relative to control, it was calculated from three independent experiments. Statistical significance was determined by unpaired two-tailed Student’s t-test using Welch’s correction with *p* < 0.05.

### 4.4. Yeast Strains and Growth Conditions

Yeast strains were grown at 37 °C in synthetic minimal media, with agar, Gal/raf with or without dropout (-uracil, -ura; -leucine, -leu; -histidine, -his) were purchased from Clontech and prepared as recommended by the manufacturer. Transformations were carried out as described in Gateway manual (Invitrogen, Carlsbad, CA, USA). Zebrafish (primers F: CACCATGGCTCGTCATGGAAAAGGC R: TTAAATTCTTTTCGACATGTATGG) Elp3 was cloned into the pYES-DEST52 destination vector (Invitrogen) using Gateway. Wild type control yeast BY4741 and ΔELP3 (kindly provided by de Crecý-Lagard) were plated in solid media in serial dilutions of 1:10 factor from left to right.

### 4.5. In Vitro Transcription, Morpholino, DNA and mRNA Injection

Capped messenger RNA (mRNA) was synthetized using either T3 or SP6 mMessage Machine (Life Technologies, Carlsbad, CA, USA) following manufacturer instructions from pENTR in which *S. Cerevisiae* was cloned using Gateway Technology (Life Technologies, Primers: F: CACC ATGGGAAAGCCAAAAAGAAGAG and R: TCAGTAAAGGTTTTTCACC). RNA probes for in situ hybridization were transcribed with either T7, SP6 or T3 RNA Polymerase (New England Biolabs, Ipswich, MA, USA) and labeled with DIG (Roche, Basel, Switzerland) following manufacturer instructions. One-cell stage eggs were microinjected with 10 nL of 0.75 mg/mL of Elp3 (translation-blocking 5′TGGCTTTCCCATCTTAGACACAAT, splicing blocking 5′CTCAAGTCACCTGACGTATAAAACAC) or control morpholino (5′ CTAACACAGATTCTACCCTTTCGGT) diluted in Danieaux buffer.

### 4.6. Protein Extraction and Western Blotting

Before protein extraction yolk was removed pipetting embryos trough a 200 µL pipette tip in ice cold PBS 10 mM PMSF and larvae disaggregated in PBS-Tween 0.3% including protease inhibitors (Thermo, Waltham, MA, USA). Proteins were resolved in 12% acrylamide gels and transferred to nitrocellulose membranes, blocked 1 h with 3% skin milk in 0.3% Tween in PBS. Incubated overnight at 4 °C in constant agitation with anti-Elp3 (Abcam, Cambridge, UK) (1:1000), secondary antibody HPR-coupled anti-rabbit IgG (1:2000) was incubated for 3 h at room temperature with gentle rocking. ECL chemiluminescence substrate was used to develop the assay (Thermo Fischer, Waltham, MA, USA).

### 4.7. Whole-Mount in Situ Hybridization, Immunofluorescence and Imaging

Embryos were fixed with 4% PFA for 2 h at room temperature. DIG-labeled probes were synthetized using either SP6/T7/T3 RNA Polymerases (New England Biolabs) and DIG-Labeling Mix (Roche). Hybridization was carried out as described [58]. Nuclei were stained with DAPI (1:200, Thermo Fischer) and F-actin with Phalloidin-TRITC conjugate (1:200, Santa Cruz Biotech, Dallas, TX, USA). For F-actin staining, larvae were fixed in 4% PFA overnight at 4 °C, permeabilized for 2 h with PBS-Triton X-100 2% at room temperature. For immunostaining, additional permeabilization was performed by partial digestion with Proteinase K (10 µg/mL) at room temperature. ZN8, F59, S58, F310 and 4D9 (Developmental Studies Hybridoma Bank) were used 1:20 and Elp3 (1:200, Abcam). Alexa Fluor-conjugated secondary antibodies were used at 1:500 (Thermo Fischer). Confocal images were captured using a Zeiss LSM 710 Meta confocal microscope (Zeiss, Oberkochen, Germany).

### 4.8. Immunofluorescence in Cross-Sections

Embryos (48 hpf) were fixed in 4% PFA, embedded in low melting point agarose-sucrose solution (1.5% agarose, 5% sucrose) and cooled on an ice pack. The embryos in agarose were kept in 30% sucrose at 4 °C until they sunk in the solution, placed in the cryostat chuck over a frozen layer of OCT compound (Tissue Tek, Breisgau, Germany) and covered with an OCT drop and frozen in liquid nitrogen. The samples were equilibrated in the cryostat overnight before sectioning. Sections of 20 μm were cut and stained.

## Figures and Tables

**Figure 1 ijms-21-00925-f001:**
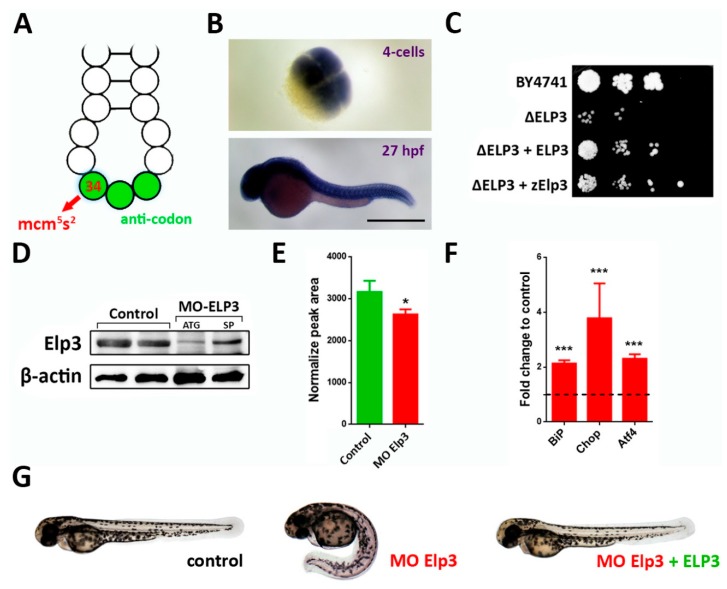
Elp3 morphants present a ventrally-curved tail. (**A**) Schematic representation of the anti-codon loop of a tRNA, indicating position 34 (red arrow) where 5-methoxycarbonylmethyl-2-thiouridine (mcm^5^s^2^U) is located. Anti-codon is depicted in green. (**B**) In situ hybridization of Elp3 mRNA in 4 cell (upper) and 27 hpf (lower) embryos. Scale bar 0.7 mm. (**C**) *Saccharomyces cerevisiae* mutants for ELP3 (ΔELP3) were transformed with an expression plasmid that contained zebrafish Elp3 (zElp3). BY4741 was used as control. Image representative of three independent experiments. (**D**) Western blot detecting Elp3 in 48 hpf samples from control and morphant embryos (MO-Elp3); β-actin was used as loading control. Samples from embryos injected with translation- (ATG) and splicing-blocking (SP). (**E**) Liquid chromatography-mass spectrometry (LC-MS) analysis of tRNA modification. Peak area values for mcm^5^s^2^U are shown normalized to the of purified tRNA for each sample (*n* = 3, * *p* < 0.05). (**F**) qPCR for UPR targets BiP, Chop and Atf4. Fold change in mRNA levels comparing morphant embryos to controls; t-test was performed for statistical analysis. *** *p* < 0.005). (**G**) Representative images of control, morphants and MO + ELP3 mRNA injected embryos by 48 hpf (images representative of at least six independent experiments, in which at least 100 embryos were analyzed).

**Figure 2 ijms-21-00925-f002:**
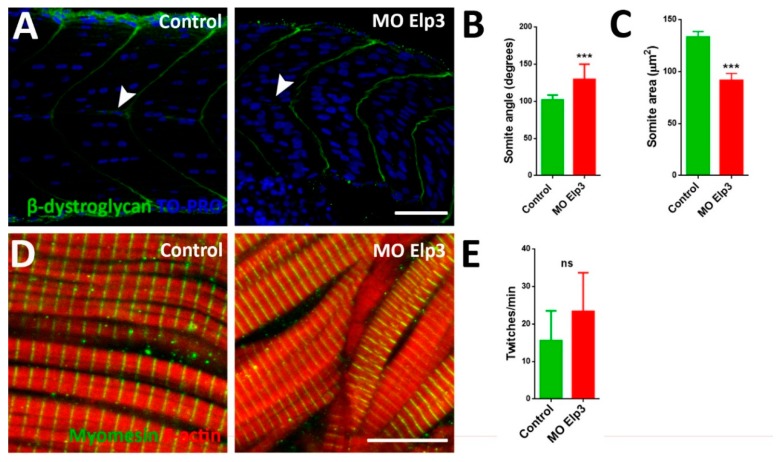
Aberrant somite shape and horizontal myoseptum in Elp3 morphants. To show somite boundary we detected β-dystroglycan in (**A**) control and morphant 48 hpf embryos, nuclei were stained with TO-PRO. Images representative of three independent experiments. Horizontal myoseptum is indicated with a white arrowhead. Scale Bar 50 µm. Also, somite angle (**B**) and area (**C**) were measured (*N* = 3, *n* = 50, t-Student. *** *p* < 0.005). Myomesin was also detected (**D**) in 48 hpf control and morphant embryos (*N* = 3, *n* = 50, t-Student). Scale bar 10 µm. (**E**) To evaluate muscle function coiling was measured counting twitches per minute in 30 hpf embryos (*N* = 3, *n* = 50, t-Student; ns non-significant).

**Figure 3 ijms-21-00925-f003:**
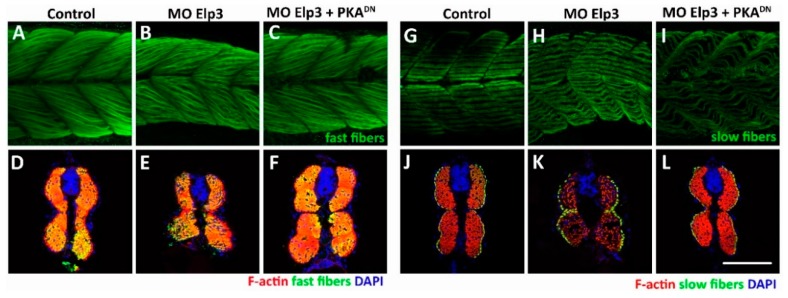
Abnormal muscle fiber morphology in morphants. Muscle fibers of control (**A**, **D**, **G** and **J**), morphants (**B**, **E**, **H** and **K**) and MO + PKADN mRNA injected (**C**, **F**, **I** and **L**) were analyzed. Upper panels show fast fibers (F59) and lower show slow fibers (S58). F-actin was stained with TRICT-phalloidin. Scale bar 50 µm. Images representative of 4 independent experiments.

**Figure 4 ijms-21-00925-f004:**
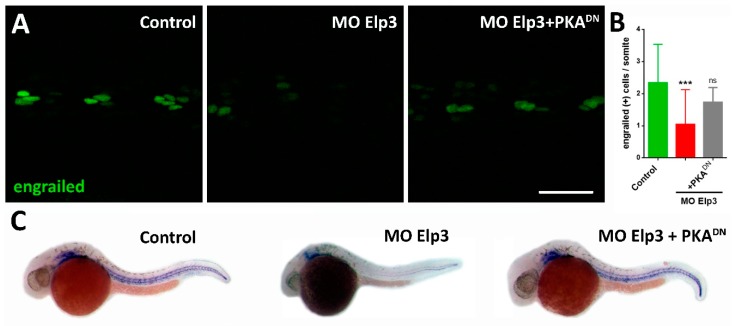
Shh activity is diminished in ELP morphants. (**A**) Immunohistochemistry usnig an anti-Engrailed antibody to visualize the adaxial cells. Control, *elp3* morphants and MO/PKA^DN^ mRNA-injected animals were analyzed 24 hpf. (**B**) The number of engrailed(+) cells were counted (*N* = 4, *n* = 30, ANOVA *** *p* < 0.005). (**C**) In situ hybridization to detect the expression pattern of col2a1 in control, *elp3* morphants and MO/PKA^DN^ mRNA-injected fish (images representative of four independent experiments). Scale bar 40 µm.

**Figure 5 ijms-21-00925-f005:**
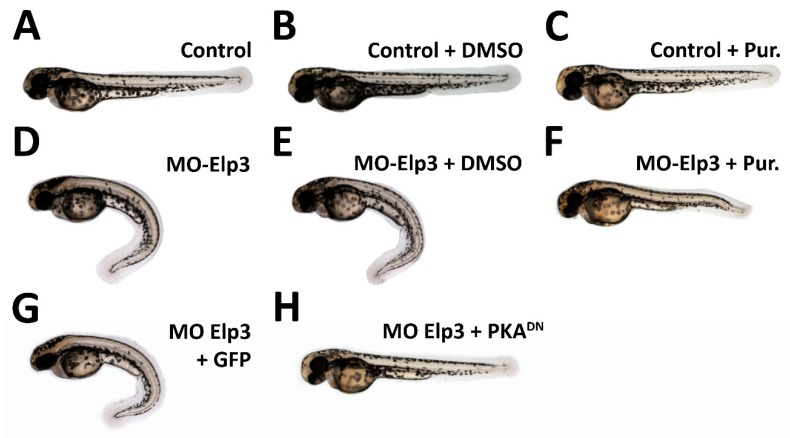
Morphant phenotype is rescued by sonic hedgehog pathway activation. Images of 2dpf larvae of (**A**) Control, (**B**) Control + 0.1% DMSO (**C**) Control + Pur 10 µM, (**D**) MO-Elp3 injected, (**E**) MO-Elp3 + 0.1% DMSO and (**F**) MO-Elp3 + Pur 10 µM. (**G**) MO Elp3 + GFP mRNA and (**H**) MO Elp3 + PKA^DN^ mRNA. Images are representative of four independent experiments.

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
