# Peer review of "Elongator Subunit 3 (Elp3) Is Required for Zebrafish Trunk Development"

_ijms, 2020, doi:10.3390/ijms21030925_

Round 1

Reviewer 1 Report

Rojas-Benitez & Allende treated zebrafish embryos with morpholinos inhibiting the expression of the elp3 gene encoding the catalytic subunit of the elongator complex. The latter is involved of the biosynthesis of the modified nucleoside mcm5s2U at position 34 of several tRNAs. Accordingly, the tRNA of morphants was shown to contain decreased amounts of mcm5s2U. In addition, morphants show aberrant, bent bodies due to disorganised muscle fibres, a phenotype reminiscent of impaired Sonic Hedgehog signalling. Indeed, the authors detected a significant downregulation of SSH activity in morphants and were able to revert the described phenotype by activating the morphants´ Sonic Hedgehog pathway.

The experiments presented by the authors seem conclusive, decently done and are well presented. Although the authors are not yet able to provide an explanation, why elp3 silencing and probably reduced mcm5s2U levels in tRNA result in the downregulation of Sonic Hedgehog pathway, the detected phenomenon is interesting and of high significance.

Typos:

Page 1, line 33: replace “… there still much to elucidate …” by “… there still remains much to elucidate …”.

Page 3, line 81: There is something wrong with the legend of Figure 1 (E). In the sentence “… are shown normalized to the of purified tRNA …” a word is obviously missing.

Page 8, line 234:

Replace “RNA was resuspended in nuclease-free water, store at -80°C until shipment …” by “RNA was redissolved in nuclease-free water, stored at -80°C until shipment …”.

Author Response

Rojas-Benitez & Allende treated zebrafish embryos with morpholinos inhibiting the expression of the elp3 gene encoding the catalytic subunit of the elongator complex. The latter is involved of the biosynthesis of the modified nucleoside mcm5s2U at position 34 of several tRNAs. Accordingly, the tRNA of morphants was shown to contain decreased amounts of mcm5s2U. In addition, morphants show aberrant, bent bodies due to disorganised muscle fibres, a phenotype reminiscent of impaired Sonic Hedgehog signalling. Indeed, the authors detected a significant downregulation of SSH activity in morphants and were able to revert the described phenotype by activating the morphants´ Sonic Hedgehog pathway.

The experiments presented by the authors seem conclusive, decently done and are well presented. Although the authors are not yet able to provide an explanation, why elp3 silencing and probably reduced mcm5s2U levels in tRNA result in the downregulation of Sonic Hedgehog pathway, the detected phenomenon is interesting and of high significance.

Typos:

Page 1, line 33: replace “… there still much to elucidate …” by “… there still remains much to elucidate …”. Page 3, line 81: There is something wrong with the legend of Figure 1 (E). In the sentence “… are shown normalized to the of purified tRNA …” a word is obviously missing. Page 8, line 234: Replace “RNA was resuspended in nuclease-free water, store at -80°C until shipment …” by “RNA was redissolved in nuclease-free water, stored at -80°C until shipment …”.

Reply:

We appreciate your review and comments. We corrected and modified as you proposed. You can see the changes as “track changes” option is activated.

Reviewer 2 Report

In the present paper, effects of knockdown of the Elongator gene ELP3 in zebrafish was analyzed. ELP3 is known in other model organisms to be essential for formation of different wobble uridine tRNA modifications and its mutation has pleiotropic effects in all cases analyzed. The paper is interesting and completes the current understanding of these effects in an additional important model organism. However, two previous studies already utilized a similar approach of ELP3 knockdown in zebrafish and results obtained in the current study are not fully consistent with the previous reports. This is discussed by the authors, but the relevant experiments are not/not fully presented (see below). Additional novel aspects of ELP3 knockdown are covered by the current study but not by the other previously published papers.

Specific points:

Line 35f: “We are interested in examining a well conserved modification, 5-methoxy-cabonyl-methyl-2-thio-uridine (mcm5s2U), which occurs at uridine in the wobble position (U34, Figure 1A) and is present in eukarya and archeae [5]”

The statement concerning archaea might be misleading. A recent paper from the Limbach group summarizes current knowledge about U34 modification in different archaea. mcm5s2U is present only in one species out of 19 analyzed (PMID: 30745370). Consider adjustment of this statement. Please correct typo in “archeae”.

The statement (line 37) that mcm5s2U is synthetized by the six-subunit Elongator Complex is too inaccurate. The thiolation (s2U in mcm5s2U) is formed by a different sulfur transfer pathway and the Elongator complex is additionally involved in ncm5U, mcm5U and ncm5Um formation (which are not mentioned at all). The latter two modifications are known to require additional non-Elongator methyltransferases. Hence, with the exception of perhaps ncm5U neither of the other modifications (including mcm5s2U) can be formed by the Elongator complex alone.

Fig.1 shows a complementation approach using Saccharomyces cerevisiae BY4741. I do not find any details in the method section about strains/plasmids/cloning procedures for this experiment. Please complete. What condition and media was used for the growth assay? In other work (for example, recent paper PMID: 31465447), loss of ELP3 results in only a very mild growth defect unless cells are subjected to heat stress. Why is a much more severe phenotype detected here? ELP3 should be in italics as far as the yeast gene is addressed.

Line 61: “To determine if the diminished Elp3 level impacted tRNA modification, we measured the levels of mcm5s2U in tRNAs from zebrafish using LC-MS (coupled liquid chromatography-mass spectrometry). In morphant animals, we detected a significant reduction in this modification compared to controls (Figure 1E), indicating a conserved molecular function for this gene in zebrafish.”

Other than the text, the legend for Figure 1 states (line 80) “Peak area values s2U are shown normalized to the of purified tRNA for each sample (n=3, p<0.005)”. Please clarify whether s2U or mcm5s2U were measured by LC-MS and complete/correct sentence. If mcm5s2U was measured and not s2U as stated, please comment on rather low reduction of modification abundance. If indeed s2U was measured this would not necessarily be consistent with loss of mcm5s2U. Commonly several different Elongator dependent modifications are measured to study loss of Elongator activity. Usually mcm5U, ncm5U and mcm5s2U are reduced/absent in cells with Elongator defects. Depending on the species analyzed, s2U may become detectable since it is formed by an Elongator independent pathway. In other words, increased rather than decreased s2U levels may be indicative of loss of Elongator function. In control animals with no ELP3 knockdown s2U should be very low or undetectable. Hence it is very important to clarify what modification was measured by LC/MS. A more complete modification abundance measurement including different Elongator dependent modifications would be desirable.

Previous studies already investigated effects of ELP3 knockdown in zebrafish (PMID: 29415125; PMID: 18996918 cited as [7]). Please comment and discuss potential differences in approach and results (see also point below). PMID: 29415125 should be relevant for the discussion of neurodegeneration since in that paper neuronal effects of ELP3 knockdown in zebrafish were analyzed.

Line 192ff: “Previous work in humans and zebrafish had shown that the loss of Elp3 leads to a shortening of motoneuron axons [7]. We examined whether this was the case in elp3 morphant zebrafish and only observed shortening in 5% of motor-neurons at 27 hpf, contrary to what Simpson et al. claimed using the same morpholinos we used [7]; also, by 48 hpf morphant showed normal motoneuron morphology (data not shown).”

Here the authors obtain results not agreeing with previously published data; however, I do not find the mentioned experiments/results to be presented in the manuscript. If such statement is made, supporting data needs to be presented rather than mentioned as “data not shown” or just presented as a summarized result.

Author Response

In the present paper, effects of knockdown of the Elongator gene ELP3 in zebrafish was analyzed. ELP3 is known in other model organisms to be essential for formation of different wobble uridine tRNA modifications and its mutation has pleiotropic effects in all cases analyzed. The paper is interesting and completes the current understanding of these effects in an additional important model organism. However, two previous studies already utilized a similar approach of ELP3 knockdown in zebrafish and results obtained in the current study are not fully consistent with the previous reports. This is discussed by the authors, but the relevant experiments are not/not fully presented (see below). Additional novel aspects of ELP3 knockdown are covered by the current study but not by the other previously published papers.

Specific points:

Line 35f: “We are interested in examining a well conserved modification, 5-methoxy-cabonyl-methyl-2-thio-uridine (mcm5s2U), which occurs at uridine in the wobble position (U34, Figure 1A) and is present in eukarya and archeae [5]”

The statement concerning archaea might be misleading. A recent paper from the Limbach group summarizes current knowledge about U34 modification in different archaea. mcm5s2U is present only in one species out of 19 analyzed (PMID: 30745370). Consider adjustment of this statement. Please correct typo in “archeae”.

The statement (line 37) that mcm5s2U is synthetized by the six-subunit Elongator Complex is too inaccurate. The thiolation (s2U in mcm5s2U) is formed by a different sulfur transfer pathway and the Elongator complex is additionally involved in ncm5U, mcm5U and ncm5Um formation (which are not mentioned at all). The latter two modifications are known to require additional non-Elongator methyltransferases. Hence, with the exception of perhaps ncm5U neither of the other modifications (including mcm5s2U) can be formed by the Elongator complex alone.

Fig.1 shows a complementation approach using Saccharomyces cerevisiae BY4741. I do not find any details in the method section about strains/plasmids/cloning procedures for this experiment. Please complete. What condition and media was used for the growth assay? In other work (for example, recent paper PMID: 31465447), loss of ELP3 results in only a very mild growth defect unless cells are subjected to heat stress. Why is a much more severe phenotype detected here? ELP3 should be in italics as far as the yeast gene is addressed.

Reply: In line 256 you can find the info about strains and growth conditions.

Our experiments were carried out at 37ºC, that why you see a more severe phenotype than in other papers. That was on purpose to show the phenotype rescues in a more visual way. Also our strain (donated by Dr. de Crecý-Lagard, in our hand showed that phenotype.

Line 61: “To determine if the diminished Elp3 level impacted tRNA modification, we measured the levels of mcm5s2U in tRNAs from zebrafish using LC-MS (coupled liquid chromatography-mass spectrometry). In morphant animals, we detected a significant reduction in this modification compared to controls (Figure 1E), indicating a conserved molecular function for this gene in zebrafish.”

Other than the text, the legend for Figure 1 states (line 80) “Peak area values s2U are shown normalized to the of purified tRNA for each sample (n=3, p<0.005)”. Please clarify whether s2U or mcm5s2U were measured by LC-MS and complete/correct sentence. If mcm5s2U was measured and not s2U as stated, please comment on rather low reduction of modification abundance. If indeed s2U was measured this would not necessarily be consistent with loss of mcm5s2U. Commonly several different Elongator dependent modifications are measured to study loss of Elongator activity. Usually mcm5U, ncm5U and mcm5s2U are reduced/absent in cells with Elongator defects. Depending on the species analyzed, s2U may become detectable since it is formed by an Elongator independent pathway. In other words, increased rather than decreased s2U levels may be indicative of loss of Elongator function. In control animals with no ELP3 knockdown s2U should be very low or undetectable. Hence it is very important to clarify what modification was measured by LC/MS. A more complete modification abundance measurement including different Elongator dependent modifications would be desirable.

Reply: It was a typing error. Now is corrected to: mcm5s2U. Find it on line 81 of the manuscript.

Morpholino is a knock-down approach, it is expected to have reduction but not absence of the modification in this condition.

We focused in mcm5s2U, that’s why we showed just the results for it. But for other elp3-dependent modification we observed similar results.

Previous studies already investigated effects of ELP3 knockdown in zebrafish (PMID: 29415125; PMID: 18996918 cited as [7]). Please comment and discuss potential differences in approach and results (see also point below). PMID: 29415125 should be relevant for the discussion of neurodegeneration since in that paper neuronal effects of ELP3 knockdown in zebrafish were analyzed.

We are aware of these papers. We were not able to reproduce the results shown in Simpson et al. and we’re confident about our results because we did CRISPR-based experiements (which we did not included in this paper because of technical problems with our fish room; all the parental mutant fish died because of problems in our equipment) which did not showed the results of Simpson et al, but were consistent with the results we show here.

On the other hand, the other paper you indicate, are overexpression experiments that modulate an ALS phenotype. I We think that when you express a tRNA-modifying enzyme in these type of situation you, at the end, you make the protein synthesis machinery stronger and processive, which allows the cell to deal with stressful situation in a better way, causing this “alleviation” of the ALS phenotype

Line 192ff: “Previous work in humans and zebrafish had shown that the loss of Elp3 leads to a shortening of motoneuron axons [7]. We examined whether this was the case in elp3 morphant zebrafish and only observed shortening in 5% of motor-neurons at 27 hpf, contrary to what Simpson et al. claimed using the same morpholinos we used [7]; also, by 48 hpf morphant showed normal motoneuron morphology (data not shown).”

Here the authors obtain results not agreeing with previously published data; however, I do not find the mentioned experiments/results to be presented in the manuscript. If such statement is made, supporting data needs to be presented rather than mentioned as “data not shown” or just presented as a summarized result.

These results are presented as a “Supplementary figure” (Figure S1)

Round 2

Reviewer 2 Report

The revised version including the addition of important experimental detail and results (Fig.S1) addressed the major points sufficiently. The yeast complementation assay (Fig.1C), however, still needs additional information: ΔELP3 + ELP3 is shown but not described anywhere. I assaume this is a complementation with the yeast gene but it is neither mentioned in the legend nor in the newly added methods. What constuct was used here? Oligonucleotide sequences used for cloning of zebrafish- and yeast ELP3 into the vectors for complementation should be added.

The recommendation to write "ELP3" in italics was not followed. I strongly recommed to do so to follow broadly accepted conventions in yeast genetics. There are also some typos that need to be eliminated.

line 11: 5-methoxy-cabonyl- should read "carbonyl"

line 36: 5-methoxy-babonyl-methy - same

line 41: Elp3 its catalytic subunit of the Elongator complex. - should read ".. is the"

Author Response

Thanks for your review.

We corrected all things you pointed out.